# Bacteria can compensate the fitness costs of amplified resistance genes via a bypass mechanism

Ankita Pal[1] & Dan I. Andersson [1]✉

Antibiotic heteroresistance is a phenotype in which a susceptible bacterial population includes a small subpopulation of cells that are more resistant than the main population. Such resistance can arise by tandem amplification of DNA regions containing resistance genes that in single copy are not sufficient to confer resistance. However, tandem amplifications often carry fitness costs, manifested as reduced growth rates. Here, we investigated if and how these fitness costs can be genetically ameliorated. We evolved four clinical isolates of three bacterial species that show heteroresistance to tobramycin, gentamicin and tetracyclines at increasing antibiotic concentrations above the minimal inhibitory concentration (MIC) of the main susceptible population. This led to a rapid enrichment of resistant cells with up to an 80-fold increase in the resistance gene copy number, an increased MIC, and severely reduced growth rates. When further evolved in the presence of antibiotic, these strains acquired compensatory resistance mutations and showed a reduction in copy number while maintaining high-level resistance. A deterministic model indicated that the loss of amplified units was driven mainly by their fitness costs and that the compensatory mutations did not affect the loss rate of the gene amplifications. Our findings suggest that heteroresistance mediated by copy number changes can facilitate and precede the evolution towards stable resistance.

The evolution and rapid spread of antibiotic-resistant bacteria is a major global health concern[1]. To combat this problem, we need improved and efficient detection and treatment strategies that can identify and prevent the emergence of resistant bacteria. While antibiotic resistance is typically considered to be a stable trait due to the presence of resistance genes acquired by horizontal gene transfer or by mutations in genes that confer resistance, several exceptions exist. Thus, resistance is sometimes unstable and only present in a small fraction of the main population, giving rise to phenotypic heterogeneity[2]. Since these bacteria are present in small fractions, they often go undetected during diagnosis and during exposure to antibiotics they can result in treatment failure.

Heteroresistance (HR) is an example of such heterogeneity and it is widely observed in various bacterial species and against several clinically relevant antibiotic classes[3–5], and it is considered to be one reason for treatment failures[6–8]. HR is defined as a phenotype wherein a main susceptible bacterial population harbors a small subpopulation of cells (typically at a frequency of $10^{-7}$ to $10^{-4}$) that are significantly more resistant to the antibiotic than the main population. Another typical characteristic of HR is that the resistant subpopulation is unstable and transient. The mechanisms that can confer heteroresistance and cause instability have been divided into two categories. The first mechanism involves frequent point mutations and even though these mutations are genetically stable they often carry a fitness cost. This cost can often be reduced by compensatory mutations, which is usually associated with loss of resistance[9], resulting in the apparent instability of the phenotype. The second and the most common mechanism of heteroresistance in Gram-negative bacteria

[1]Department of Medical Biochemistry and Microbiology, Uppsala University, Box 582 SE-751 23, Uppsala, Sweden. ✉e-mail: dan.andersson@imbim.uu.se

involves tandem gene amplification of resistance genes that in single-copy are not sufficiently active to confer resistance, but when amplified to higher copy numbers the cells become resistant[9–12]. Gene amplification-mediated heteroresistance is known to arise at high frequency by homologous recombination between directly repeated sequences (e.g. insertion sequences, ribosomal RNA operons, and transposons) that flank the resistance genes and are present in the chromosome and/or plasmids. In the absence of selection, these gene amplifications are easily lost since the amplified regions can be removed by unequal crossing over events between the direct repeat sequences. This loss and reversion to susceptibility is thought to be driven by both the mechanistic loss rate and the fitness costs of the amplification[13].

In order for these amplifications to be stably maintained in the population long-term the fitness cost needs to be ameliorated. Such compensation mechanisms are well-characterized for several other types of antibiotic resistance mutations, and include, for example, intra- and extragenic point mutations that restore functionality to an impaired target (e.g. ribosome, RNA polymerase etc.)[14–18]. However, the genetic mechanisms underlying compensation in gene amplification-mediated heteroresistant strains remains unknown. Given the dynamic nature and instability of gene amplifications, the compensation of fitness cost could progress in several different ways, including: (i) *restructuring* of the amplified regions in which deletions remove the costly genes within the amplified region[19,20], (ii) *silencing* of costly genes by mutations in genes that regulate the costly genes or a general silencing by H-NS like protein[21], and (iii) *replacing* the costly amplifications by other less costly mutations that bypass the need for high-level amplifications. Compensatory mechanisms to reduce the fitness costs can involve one or more of the above possibilities.

In this study, we examined how the fitness cost conferred by gene amplifications can be compensated in four clinical heteroresistant isolates of Gram-negative bacteria. Results show that fast emergence of highly resistant bacteria carrying large amplifications occur when exposed to antibiotics. The gene amplifications are associated with extensive fitness costs, and compensation to reduce these costs takes place rapidly via a myriad of low-cost resistance mutations located in the bacterial chromosome that reduce the need for high-level gene amplifications. Furthermore, analysis of the loss dynamics shows that the main driving force for loss is indeed the fitness costs of the amplifications and that the compensatory chromosomal mutations do not alter the stability of the gene amplifications in the absence of antibiotics. These results enhance our understanding of the dynamics of heteroresistance and demonstrate a new pathway for compensation of the fitness costs associated with increased gene copy numbers.

## Results

### Isolation of mutants at higher levels of antibiotic
To enrich mutants with higher levels of gene amplifications and fitness costs, we used four clinical isolates of *E. coli* (DA33135 and DA33137), *K. pneumoniae* (DA33140) and *S. enterica var*. Typhimurium LT2 (DA34827) (designated *S.* Typhimurium throughout). These strains had been previously shown to be HR due to tandem gene amplifications of the resistance gene present on the plasmid except for the strain DA34827 wherein the resistance gene was located on the chromosome. Table 1 shows the type of resistance gene, the size of the amplified regions and the location of the resistance gene within the amplified unit[9]. The strains were HR to tobramycin, (DA33135), gentamycin (DA33137, DA33140), and tetracycline (DA34827). Enrichment of mutants with amplifications was carried out by serially streaking single colonies of each strain on agar plates containing an increasing concentration of the respective antibiotics (1X, 4X, 16X, and 24X MIC of the main population). For each of the strains, four lineages were evolved independently (Fig. 1A).

### Increased gene amplification in the isolates reduces fitness and increases resistance levels
Increases in resistance gene copy number were measured using digital droplet PCR (ddPCR) for each of the selected mutants at the corresponding antibiotic concentration at which they were isolated. For all the isolated mutants, there was a rapid increase in the gene copy number which was correlated with the antibiotic concentration (Fig. 2A). The increase in the copy number was the fastest and also highest for the strains DA33137 and DA33140, and at 24X MIC all the isolated mutants showed a 20- to 80-fold increase in their copy numbers as compared to their parental strains. Thus, for the heteroresistant strains carrying the resistance gene on a plasmid, the copy number of the resistant gene in the mutants isolated from 24X MIC of the antibiotic was ~80, whereas for the heteroresistant strain carrying the resistance gene on the chromosome, the copy number was ~20. Table 1 shows the size of the amplified units and the type and approximate location of the resistance gene within the amplified unit.

We estimated the fitness cost associated with increased copy number for each of the mutants at the corresponding antibiotic concentration at which they were isolated by measuring the exponential growth rate (Supplementary Fig. 1). We then calculated the relative fitness of the mutants containing increased copy number of the resistance gene by comparing mutant fitness to that of the corresponding wild-type strain containing only a single copy of the resistance gene. For all the mutants isolated at 1X and 4X we did not observe any significant decrease in fitness despite these strains showing a substantial increase in the copy number of the resistant gene as compared to their respective wild-type strains. However, there was a significant decrease ($p < 0.01$, Student's $t$ test) in the relative fitness at 16X and 24X MIC with all the mutants having a relative fitness of ~60% as compared to the parental strains at the end of the experiment (Fig. 2B, and C).

We further tested whether the increase in the gene copy number at each of the antibiotic concentration was associated with increased resistance levels by measuring the MIC of the relevant antibiotics by Etest. For all the isolates, we observed a high increase in the MIC with most of the mutants selected at 4X, 16X, and 24X MIC showing a MIC >256 mg/l (Supplementary Table 1). Thus, evolving the strains at increasing concentrations of antibiotics not only increased the copy number of the resistance gene but also increased the associated fitness cost and the resistance level in all isolated mutants.

### Fitness of bacteria carrying costly gene amplifications can be rapidly increased during serial passage in presence of antibiotic
To investigate how the fitness cost in the mutants can be genetically ameliorated, we carried out a compensatory evolution experiment with all mutants isolated at 24X MIC (Fig. 1B). Each of the sixteen mutants (four from each of the four strains) were evolved in three parallel lineages by serially passaging them with 1:200 dilutions in a 96-well plate containing 200 ml of Mueller-Hinton (MH) broth with 24X MIC of the respective antibiotic (Fig. 1B). The serial passages were performed every 24 hours and were carried out for 100 generations, after which single clones were isolated by plating them on agar plates containing 24X MIC of the corresponding antibiotic. These clones were used for all further experiments.

The extent of the compensation of the fitness cost was determined by measuring the exponential growth of the evolved single clones isolated from the compensatory evolution experiment (Supplementary Fig. 2) and comparing them to that of the wild-type strains (Fig. 3). We observed that most of the mutants had completely or partially compensated the fitness cost ($p < 0.01$, Student's $t$ test) after 100 generations with several mutants having growth rates similar to that of the corresponding wild-type strain.

Next, we determined the levels of gene amplifications in all evolved strains from the endpoint of the compensatory evolution

**Table 1 | Heteroresistant strains used for antibiotic selection**

| Strain | Species | Antibiotic | Amplified region (kb) and type and location of resistance gene (arrow) | Location |
|---|---|---|---|---|
| DA33135 | *E. coli* | Tobramycin | 27.7 *aac(3)-IId* | Plasmid |
| DA33137 | *E. coli* | Gentamicin | 20.7 *aac(3)-IId* | Plasmid |
| DA33140 | *K. pneumoniae* | Gentamicin | 4.2 *aac(3)-IIa* | Plasmid |
| DA34827 | *S.* Typhimurium | Tetracycline | 9.7 *tet(A)* | Chromosome |

The name of the resistance gene has been shown in italics.

experiment. We observed that the majority of the mutants had a significant reduction in the resistance gene copy number (Fig. 3). Furthermore, upon measuring the resistance levels of the evolved strains, we observed that the isolated clones still had very high levels of resistance (>256 mg/l) despite the reduction in the gene amplifications and the increase in growth rate (Supplementary Table 2). One explanation for this finding is that these mutants likely acquired other low-cost resistance mutations that partially bypass the need for high-level amplifications (i.e., replacement model 3 in the Introduction). In other words, the high-level resistance is likely caused by a combination of moderate amplification and other resistance mutations.

**Many types of mutations can bypass the need for high-level amplification**

To identify the resistance mutations that might relieve the need for high-level amplification and thereby confer the growth rate compensation, we whole genome sequenced all of the evolved clones from the compensatory evolution experiment. We sequenced 12 mutants for each of the strains from the compensatory evolution experiment giving a total of 48 mutants. Additionally, we also whole genome sequenced the high copy number parental isolates obtained at 24X MIC to provide comparative control sequences to help us identify the compensatory mutations that appeared during evolution in presence of antibiotic.

For the mutants sequenced after compensatory evolution, a majority of the identified mutations were located in the bacterial chromosome (Supplementary Table 3). They were predominantly either known resistance mutations or mutations arising due to media adaptation. We also observed a few mutations whose role in conferring resistance was unknown. Across all the mutants, the patterns of mutations acquired were unique indicating that many different evolutionary paths exist to compensate for the growth rate defect.

For the *E. coli* mutants that were evolved on tobramycin, mutations were predominantly observed in the genes such as *cpxA*, *nuoG* and *rluD*, which are known to confer resistance to various antibiotics, including aminoglycosides[22–25]. Additionally, several mutations were also found in the gene *lrhA* as well as in the intergenic region between *fimA* and *fimE*. These genes are associated with fimbriae formation and regulation and are known to be frequently mutated during media adaptation[26]. Several mutations were also observed in genes such as

*qorB*, *pitA*, *bglB*, *pgsA*, *yiaO*, and *gspK* whose role in resistance is not clear. Similarly, for the *E. coli* mutants that were evolved on gentamycin, mutations were mostly found in known resistance genes like *cpxAR*[23], *cyoB*[22], and *rsmF*[27]. Similar to the mutants evolved on tobramycin, the mutants evolved in gentamicin also contained common media adaptation mutations in genes such as *fimE* and structural variants in genes *lrhA* and the intergenic region between *fimA* and *fimE*. Interestingly, in the *K. pneumoniae* mutants evolved on gentamycin, we did not observe mutations arising due to media adaptations. Instead, mutations were mostly observed in already known resistance genes such as *cpxA*[23], *nuoA*[28], and *arcB*[29]. Additionally, mutations were observed in genes like *wcaJ*, *lpxM*, and *iroE* whose roles in resistance development are yet to be deciphered. In case of the *S.* Typhimurium mutants evolved on tetracycline, a majority of the mutations were in the *tetR/acrR* gene and in the intergenic region between *tetR/acrR* and *ramA* gene. The *tetR/acrR* genes encode transcriptional repressors and are known to confer resistance to various antibiotics[30]. Also, in 2 out of the 12 mutants evolved on tetracycline, mutations were observed in the *rrf* gene that codes for ribosome recycling factor and possibly confers antibiotic resistance in these mutants.

Apart from these mutations, the compensated mutants as well as the high copy number parental isolates showed amplification of the resistance gene as deciphered by comparing the coverage of the mapped sequencing reads. The pattern of amplification levels agreed well with the copy number changes measured using ddPCR. The parental isolates showed no other mutations apart from these amplifications.

To validate that these secondary mutations acquired in the mutants as a result of compensatory evolution indeed confer resistance, 4 mutations were reconstructed (the *nuoG* and *cyoB*, mutations in *E. coli* and the *tetR/acrR, and rcsB* mutation in *S.* Typhimurium). Since the used clinical strains are not amenable to genetic manipulation via λ-red recombineering, we tested the effects of these mutations in lab strains of *E. coli* (MG1655) and *S.* Typhimurium. The MIC in these reconstructed strains were then tested (*nuoG* on tobramycin, *cyoB* on gentamicin and, *tetR/acrR*, and *rcsB* on tetracycline). We found that all of these 4 mutations resulted in an increased resistance (Supplementary Table 4), suggesting that they can bypass the need for the costly amplifications and thereby act to increase the growth rate in presence of antibiotic. To further support this hypothesis, we measured the

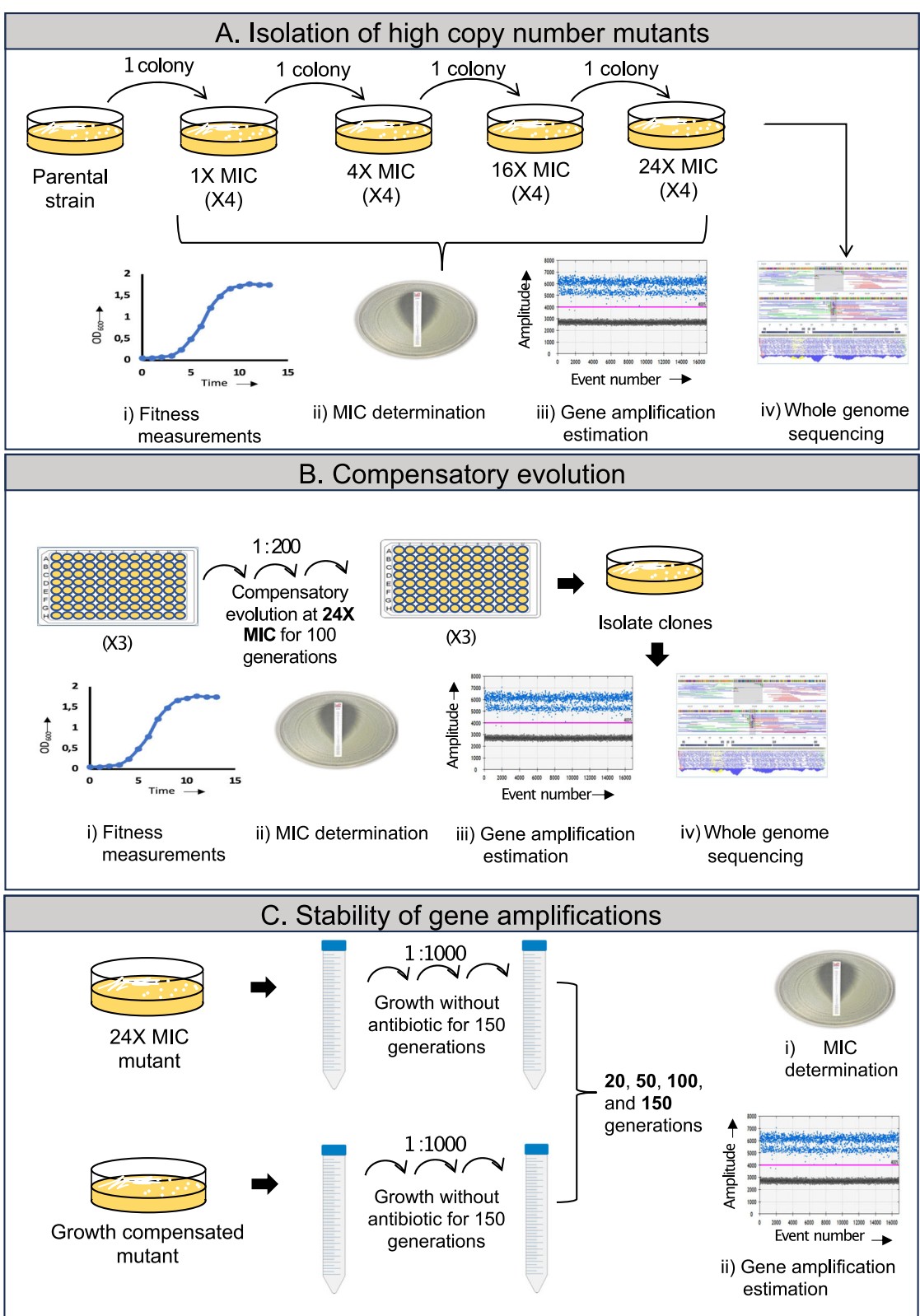

growth rates of the four reconstituted mutants. As expected, the fitness costs of these mutations were lower (relative growth rate ~0.96-0.98, Supplementary Table 4) ($p < 0.01$, Student's $t$ test) as compared to a high-level amplification (relative growth rate ~0.6, Fig. 2C). These data demonstrates that these new mutations can provide resistance at a lower cost (2–4%) than the original high-level gene amplifications (40%, see Fig. 2C).

**Loss rates of gene amplifications in compensated mutants in absence of selection are unaffected by chromosomal resistance mutations**

Tandem gene amplifications are intrinsically unstable and during growth in the absence of antibiotic pressure they are lost by homologous recombination between sister chromatids. To measure the loss rates of gene amplifications and to determine whether the compensatory

**Fig. 1 | Schematic representation of the main methods used in this study.**
**A** Selection of resistant bacteria with increased copy number and fitness cost by
serially streaking 4 heteroresistant strains at progressively higher concentrations of
their respective antibiotic in 4 parallel replicates. The isolated mutants were
characterized by measuring the fitness, MIC, and resistance gene copy number.
Mutants isolated at 24X MIC exhibited the highest fitness cost and highest levels of
gene amplifications and were whole genome sequenced and used for further stu-
dies. **B** Compensatory evolution was performed at 24X MIC antibiotic

concentration with the mutants isolated at 24X MIC in 3 replicates for each of the 16
mutants isolated previously. After serial passaging for 100 generations, the mutants
were characterized by measuring fitness, MIC, resistance gene copy number and
whole genome sequencing. **C** The stability of the gene amplifications was measured
in the compensated mutants and was compared to its respective parent isolated at
24X MIC. This was done by serially passaging these strains in antibiotic-free media
for 150 generations and then measuring the changes in MIC and resistance gene
copy number.

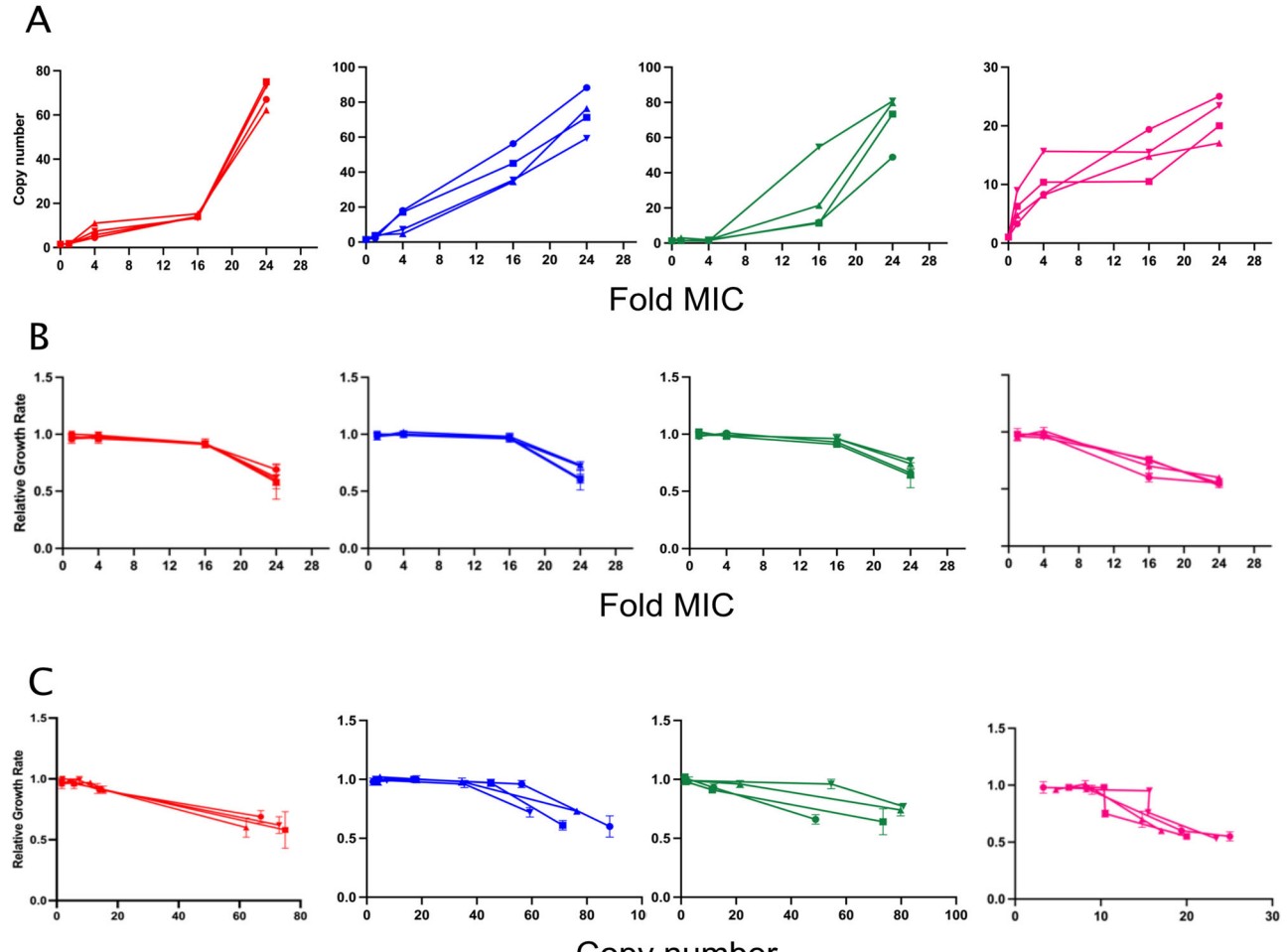

**Fig. 2 | Characterization of resistant mutants. A** Changes in copy number of the
resistance genes with increasing antibiotic concentration. **B** Relative growth rates
with increasing antibiotic concentration. **C** Changes in the relative growth rate with
increasing copy number of the resistance genes. Red represents the independent
mutants isolated from DA33135 at each concentration of tobramycin tested
(DA76789, DA76790, DA76791, DA76792 at 1X MIC; DA76805, DA76806, DA76807,
DA76808 at 4X MIC; DA76821, DA76822, DA76823, DA76824 at 16X MIC; DA76837,
DA76838, DA76839, DA76840 at 24X MIC), blue represents the independent
mutants isolated from DA33137 at each concentration of gentamicin tested
(DA76793, DA76794, DA76795, DA76796 at 1X MIC; DA76809, DA76810, DA76811,
DA76812 at 4X MIC; DA76825, DA76826, DA76827, DA76828 at 16X MIC; DA76841,
DA76842, DA76843, DA76844 at 24X MIC), green represents the independent

mutants isolated from DA33140 at each concentration of gentamicin tested
(DA76797, DA76798, DA76799, DA76800 at 1X MIC; DA76813, DA76814, DA76815,
DA76816 at 4X MIC; DA76829, DA76830, DA76831, DA76832 at 16X MIC; DA76845,
DA76846, DA76847, DA76848 at 24X MIC) and pink represents the independent
mutants isolated from DA34827 at each concentration of tetracycline tested
(DA76801, DA76802, DA76803, DA76804 at 1X MIC; DA76817, DA768118, DA76819,
DA76820 at 4X MIC; DA76833, DA76834, DA76835, DA76836 at 16X MIC; DA76849,
DA76850, DA76851, DA76852 at 24X MIC). Copy number was measured from the
cell population for 3 independent lineages (biological replicates) and 1 technical
replicate. Relative growth rate was measured from 6 biological replicates. Data are
presented as mean values ± standard deviation indicated by error bars. Source data
are provided as a Source Data file.

mutations affect this rate, we performed a serial passage experiment
with the compensated mutants obtained after evolution at 24X MIC
(Fig. 1C). The experiment was performed for 150 generations of growth
in the absence of antibiotic by serially passaging 2 μl of culture to 2 ml of
MH media once daily following overnight growth to full density (1000-
fold dilution per passage, corresponding to approximately 10 genera-
tions of growth). Copy number was measured after 20, 50, 100, and 150

generations of growth and compared with the copy number obtained
from a similar serial passage experiment performed with the parental
isolates obtained at 24X MIC to determine if the loss dynamics of the
amplification cassettes were affected by the mutations acquired during
compensatory evolution.

For all mutants, we observed a rapid loss of the amplified units
with some variability. The amplification units were lost faster during

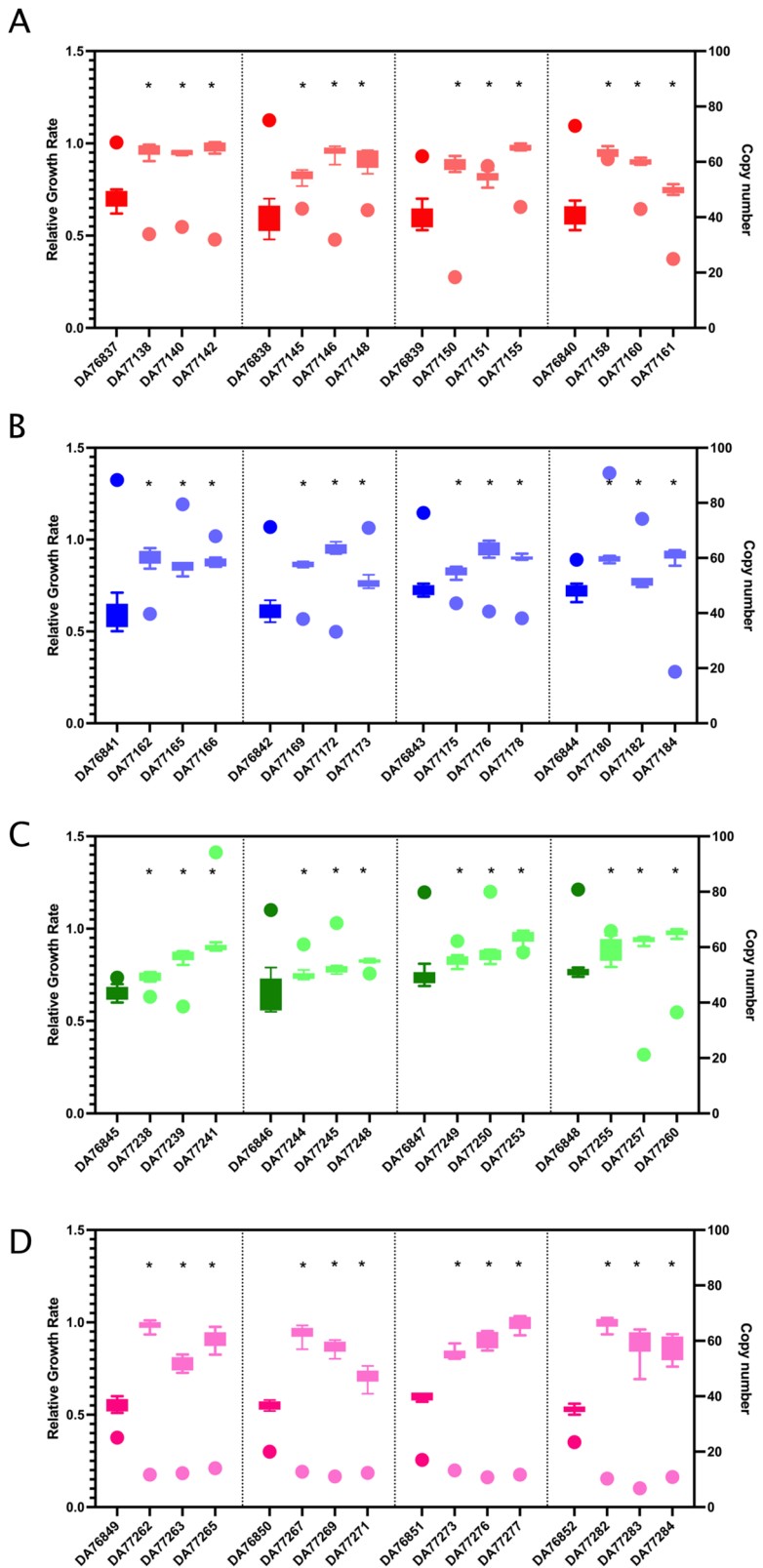

the first 50 generations of growth and the rate slowly diminished as the copy number was reduced (Fig. 4). Since a similar pattern of gene amplification loss were seen in both the parental mutants selected at 24X MIC (Supplementary Fig. 3) and the compensated mutants (Fig. 4), it is clear that the chromosomal mutations that were acquired during compensatory evolution plays no role in the stability of these gene amplifications, and in the absence of selection they are lost from the mutant strains with the same rate as the mutants lacking these chromosomal mutations.

## Modelling based on determined parameter values

The mechanism of amplification inherently involves homologous recombination that either leads to enrichment of amplification units or their loss based on the selection pressure. From the results of our

**Fig. 3 | Relative growth rate and copy number of the compensated mutants and their respective parental strains. A** Compensated mutants of *E. coli* isolated on 24X MIC tobramycin after 100 generations (red). **B** Compensated mutants of *E. coli* isolated on 24X MIC gentamicin after 100 generations (blue). **C** Compensated mutants of *K. pneumoniae* isolated on 24X MIC gentamicin after 100 generations (green). **D** Compensated mutants of *S. Typhimurium* isolated on 24X MIC tetracycline after 100 generations (pink). The darker shade represents the high copy number parental strains isolated at 24X MIC and the lighter shades represent its three independent lineages for compensatory evolution experiment. Copy number (circles) was measured from the cell population for three independent lineages

(biological replicates) and one technical replicate. Relative growth rate (box plots) was measured from six biological replicates. Data are presented as mean values ± standard deviation. The asterisks (*) indicate statistically significant changes in growth rate in the compensated mutants compared to their respective parental strains. The statistical testing was done using unpaired two-tailed Student's *t* test. Exact P values are given in Source data file. For the box-plots, minima and maxima are the 25th and 75th percentiles respectively and the center is 50th percentile or the median value. The whiskers extend from the smallest to the largest value in the group. Source data are provided as a Source Data file.

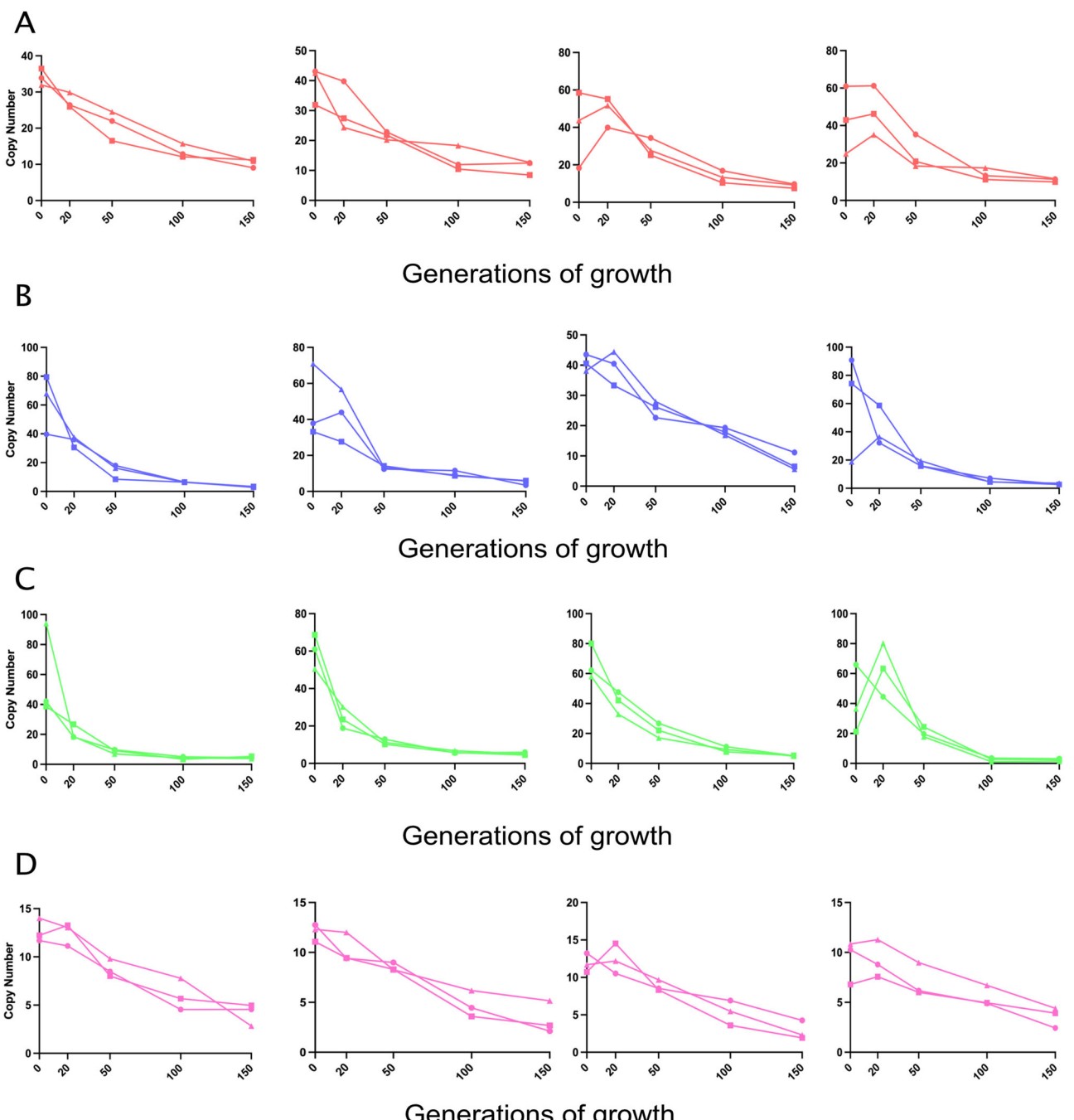

**Fig. 4 | Loss dynamics of the gene amplifications with time in absence of antibiotics.** For each species and antibiotic combination three independent compensated mutants were examined for their loss rates using three biological replicates. **A** Compensated mutants of *E. coli* isolated on tobramycin.

**B** Compensated mutants of *E. coli* isolated on gentamicin. **C** Compensated mutants of *K. pneumoniae* isolated on gentamicin. **D** Compensated mutants of *S. Typhimurium* isolated on tetracycline. Source data are provided as a Source Data file.

experiment, we observed rapid loss of gene amplifications when the mutants were grown for several generations in the absence of antibiotic. To understand the dynamics of this loss of the amplified units, we used a previously described deterministic model of homologous recombination[31,32]. This model was used to theoretically determine the recombination rates ($k_{rec}$) and the physiological cost ($s$) of carrying each extra gene copies, that best mimicked the loss dynamics from our experimentally measured data (Supplementary Fig 4). The inferred parameter values agreed very well with the experimentally observed decline in array size (Supplementary Table 5). We did not observe any changes in the kinetic parameters between strains carrying similar number of gene copies when comparing the compensated mutants with their parental strains isolated at 24X MIC which indicated that the kinetic parameters were not affected by the chromosomal mutations acquired during compensatory evolution. The loss rate of an amplification will be determined by both the mechanistic loss rate ($k_{rec}$) and the fitness cost ($s$) of the amplified state[33]. In our analysis, the values for $k_{rec}$ varied between $3.5 \times 10^{-5}$ and $2 \times 10^{-2}$ with a mean and median values of 0.001 and 0.0003, respectively; the values for $s$ (cost per extra copy in the amplified array) varied between $3 \times 10^{-3}$ and $2.5 \times 10^{-2}$ with mean and median values of −0.009 and −0.008, respectively. In Fig. 5, the values for $k_{rec}$ and $s$ are plotted for each strain and compensated lineage. We measured the goodness-of-fit of the model to our experimental data. In majority of the cases, we observed good fit ($R^2 > 0.9$, $p < 0.01$) of the theoretical estimation to our experimental data. This indicates that the model provides a good framework to predict recombination rates and fitness costs with high reliability

(Supplementary Fig 4). We also found that for the majority of cases, $s > k_{rec}$, meaning that the loss of the amplifications in absence of selection is mainly driven by the fitness costs of the amplifications (i.e. the fitness difference between cells with different copy numbers) rather than by the mechanistic loss rate (Fig. 5). Overall, the high recombination rates and fitness costs indicates the inherent instability associated with gene amplifications in the absence of antibiotics that in turn impairs their detection in the population. This model provides a quantitative framework to determine the importance of fitness costs and recombination rate on the stability of gene amplifications.

## Discussion

Heteroresistance has been an area of active research since it was first described in the 1940s[5,9,34]. However, the genetic basis underlying the emergence and maintenance of the heteroresistance phenotype has until recently been poorly understood. In Gram-negative bacteria, the heteroresistance phenotype is mediated by tandem amplification of the antibiotic resistance gene that apart from being unstable are also known to carry a fitness cost[9,13].

In order to understand and predict the evolution of antibiotic resistance, it is crucial to understand the compensatory strategies that bacteria can use to ameliorate the fitness costs associated with gain of resistance. Several studies have identified a plethora of mechanisms by which bacteria can reduce the costs associated with mutations in antibiotic targets[14–17,35–38], efflux/uptake mutations[39,40], and carriage of plasmids and resistance genes[18,41,42]. Until this study, the compensatory mechanisms that ameliorate the fitness cost

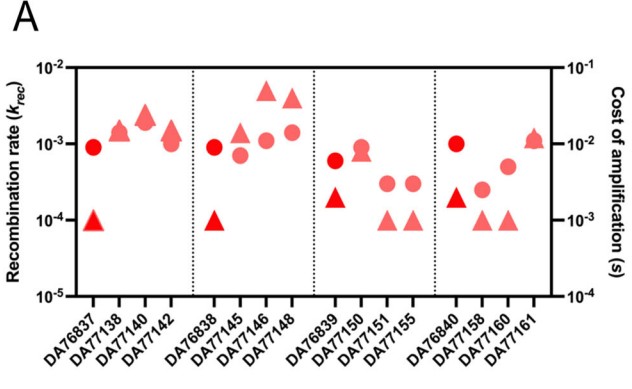

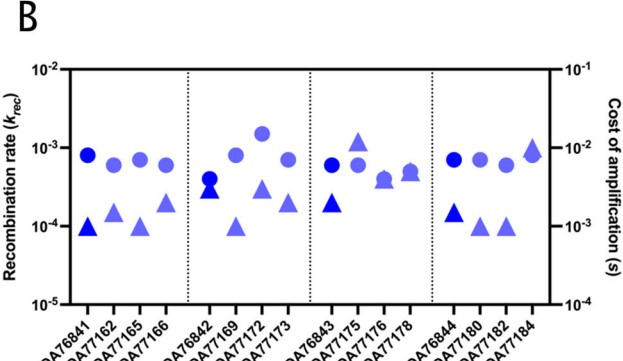

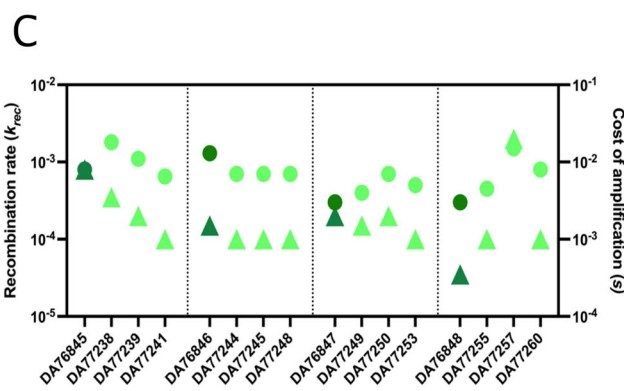

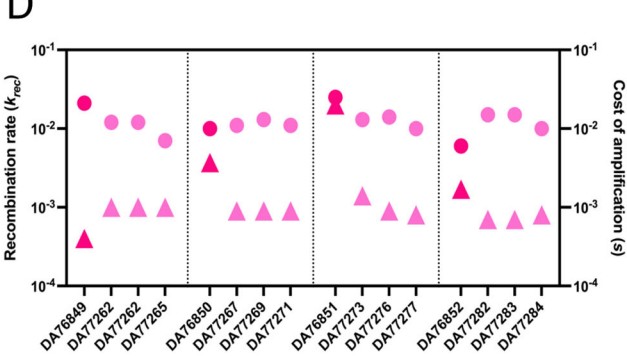

**Fig. 5 | Recombination rates (krec) and s-values for resistant and compensated strains.** Absolute values of $k_{rec}$ (triangles; left y-axis) and s-values (circles; right y-axis). **A** Compensated mutants of *E. coli* isolated on 24X MIC tobramycin after 100 generations (red). **B** Compensated mutants of *E. coli* isolated on 24X MIC gentamicin after 100 generations (blue). **C** Compensated mutants of *K. pneumoniae* isolated on 24X MIC gentamicin after 100 generations (green). **D** Compensated

mutants of *S. Typhimurium* isolated on 24X MIC tetracycline after 100 generations (pink). The darker shade represents the high copy number parental strains isolated at 24X MIC and the lighter shades represent the three independent lineages from the compensatory evolution experiment. Source data are provided as a Source Data file.

associated with gene amplifications-mediated heteroresistance were unknown.

Our findings show that heteroresistant populations with high resistance mediated by high levels of gene amplifications are enriched rapidly when selected at increasing concentrations of antibiotics (Fig. 6). Interestingly, for some cases the fitness cost did not linearly correlate with an increase in the gene amplifications. For instance, no significant fitness cost was observed in several mutants even when they carried up to ~40 copies of the amplified units, but with higher copy number (~80 copies) these amplifications resulted in a substantial fitness reduction (~40% when compared to their parental isolates) (Fig. 2C, strain DA33137 on gentamicin and strain DA34827 on tetracycline). These findings suggest that there is, for presently unknown reasons, some sort of threshold effect.

The mutants selected at 24X MIC showed both high fitness costs and high levels of gene amplifications and were therefore chosen for the compensatory evolution experiments to identify the mechanisms responsible for the amelioration of fitness costs. The evolution results show that the fitness cost can be compensated rapidly within 100 generations of growth while still maintaining the high resistance levels in almost all of the tested lineages (>256 mg/l). With regard to the three proposed models (see Introduction) for how compensation could have occurred –*restructuring, silencing,* and *replacing,* sequence analyses suggest that partial replacement of the gene amplification by other less costly mutations is the main mechanism by which fitness is increased. Thus, a majority of the compensated mutants (94%) showed an approximately 50% reduction in copy number while concomitantly acquiring additional mutations, mostly

within known resistance genes, that alleviated the need for high-level amplification (Fig. 6).

At the onset of this work, we also expected to find examples of chromosomal restructuring, such as the loss of costly genes in amplified units by deletions[19,20]. It is likely that this mechanism was not observed in this study because of the high copy numbers present in our strains (approximately 50-80 copies). A deletion in a single copy in a highly amplified array would be expected to confer only a very small fitness increase and deletion of the costly genes in all amplified units would require a lengthy stepwise process in which each step is associated with a small fitness gain. Conversely, in strains with lower amplification levels (e.g. a duplication), a deletion of a costly gene in one of the copies will have a relatively larger fitness-increasing effect and can be rapidly selected[19,20].

The unstable and transient nature of gene amplifications make it challenging to detect and analyze heteroresistant isolates. Since the stability and fitness costs of the gene amplifications determines the maintenance of the resistance phenotype in the population, it is important to determine the recombination rates and fitness costs, and whether compensatory mutations affect them. Using a previously developed deterministic model for homologous amplification we quantified the recombination rate ($k_{rec}$) and cost ($s$) of carrying extra copies of DNA[31]. By combining this model with our experimentally derived loss rates obtained from bacteria grown in the absence of antibiotics, we determined the range of parameter values that best fitted the experimental data. The $k_{rec}$ varied between approximately $10^{-4}$ to $10^{-2}$ per cell and generation whereas the $s$-values (the fitness reduction conferred by each amplified unit) were approximately 1 to 2

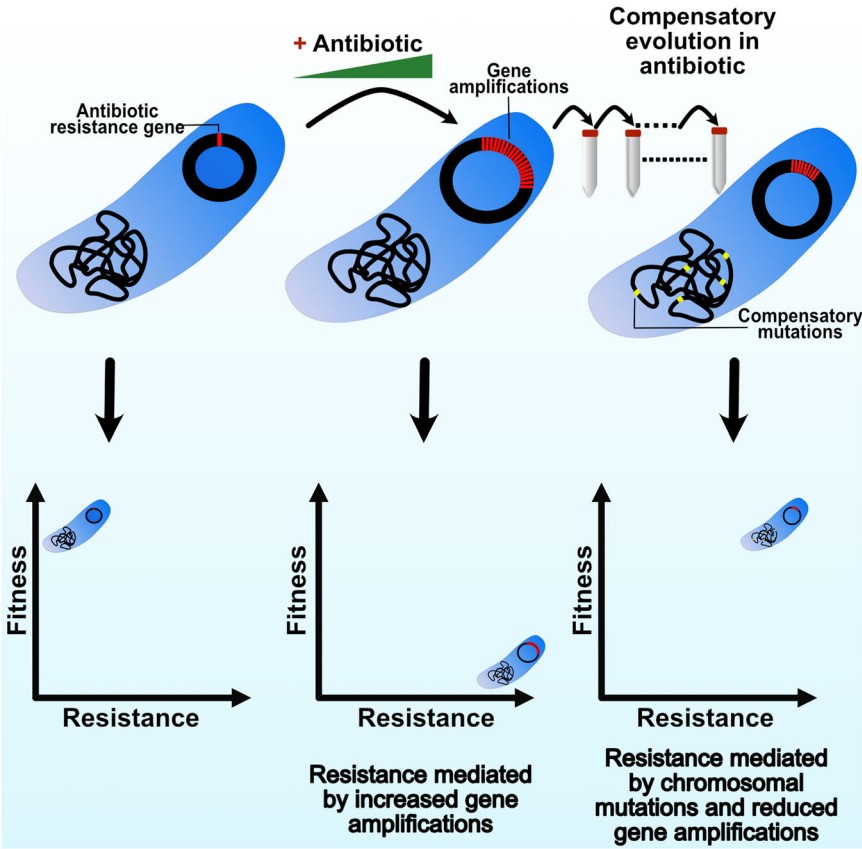

**Fig. 6 | Model for the compensation of gene amplification-mediated fitness cost.** Exposure to increasing concentrations of antibiotics leads to rapid high-level amplification of the antibiotic resistance gene. The increased copy number increases the resistance level and strongly reduces the fitness of the bacterium. Subsequent adaptive evolution at a continuous high antibiotic concentration (24X MIC) resulted in compensation of the fitness cost via secondary chromosomal resistance mutations that mitigate the need for high level gene amplification to achieve high-level resistance.

orders of magnitude higher (Fig. 5 and Supplementary Table 5). The high degree of fit observed between the model and experimental data suggest that this model provides a good framework to determine recombination rates and fitness costs with reasonable accuracy. Importantly, these data demonstrate that the main factor determining the loss rate of the tandem arrays is the fitness costs of the amplified units rather than the recombination rate. Finally, the loss rates of the original amplified and growth-compensated strains were not systematically different, showing that the additional chromosomal mutations do not alter the stability of the gene amplifications. The high rates of recombination as well as the high fitness cost associated with gene amplifications show the ease with which they can be lost from the population thereby making them difficult to detect and treat under clinical conditions.

Overall, our study shows that the rapid development of high-level resistance in heteroresistant populations in response to antibiotic selection is primarily due to selection for tandem gene amplification of resistance genes. Specifically, these amplifications confer a fitness cost that can be rapidly ameliorated by second-site resistance mutations that bypass the need for high-level amplification. Our findings are especially important for clinical settings, as compensatory mechanisms that reduce the high fitness effects may promote development of stably resistant bacteria arising from heteroresistant cultures[43], and illustrate the need for better detection and surveillance methods as well as treatments to more efficiently identify and eradicate hetero-resistant bacteria[6]. Finally, the process observed here is likely to occur also during adaptation to selective conditions other than antibiotics. Rapid increases in gene copy number via tandem amplifications can, for example, be observed also during host adaptation, carbon starvation and other stressful conditions[44,45], and compensatory mutations that alleviate the need for costly high-level amplifications could subsequently stabilize these bacteria in the population.

## Methods

### Bacterial strains, growth conditions, and antibiotics
All bacterial isolates used in this study are listed in Supplementary Table 6. For all the evolution and serial passaging experiments, Mueller-Hinton medium (MH) (Becton Dickinson, MD, USA) was used either in broth or in agar plates. For the strain constructions, LB broth (Sigma-Aldrich) was used as either broth or agar plates. The antibiotics used in this study (tobramycin, gentamicin, and tetracycline) were purchased from Sigma-Aldrich. Unless otherwise mentioned, all cultivations were performed at 37 °C with liquid cultures shaken at 190 rpm for aeration.

### Isolation of resistant mutants with gene amplifications and reduced growth rates
To isolate mutants with increased levels of gene amplifications and reduced growth, 4 clinical isolates heteroresistant to different antibiotics were used. For isolation of mutants, a single colony from each of the four different strains was sequentially streaked on MH agar plates containing increasing concentrations of the antibiotic (1X, 4X, 16X and 24X MIC of the main population) for which it was shown to be heteroresistant. A total of 4 independent lineages were maintained for each of the 4 strains and the streaking was done every 24 hours. A single colony isolated from every antibiotic concentration was grown overnight in Mueller-Hinton broth supplemented with the same antibiotic concentration at which it was isolated. An aliquot from that culture was stored at −80 °C in 10% dimethylsulfoxide (DMSO). In parallel, from the same overnight culture, an aliquot of 340 µl was pelleted and stored at −20 °C for DNA isolation to determine the levels of gene amplifications by digital droplet PCR (ddPCR). Additionally, an aliquot of the same culture was also used for MIC determination using Etests. For the mutants isolated at 24X MIC, an aliquot of 500 µl was pelleted and stored at −20 °C for DNA isolation and subsequent whole genome sequencing (WGS).

### Compensatory evolution of heteroresistant population
The compensatory evolution experiment was performed with three lineages for each isolated mutants (3 × 4) demonstrating high copy number and reduced growth rates for each of the strains. The evolution experiment was carried out for 100 generations in 96-well plates containing 200 µl of MH broth supplemented with 24X MIC of the respective antibiotic. The cell cultures were grown at 37 °C with shaking at 190 rpm and 1 µl culture was serially passaged into fresh media once daily. Single colonies were isolated from cells grown in the 96-well plates and were used to inoculate MH broth supplemented with 24X MIC antibiotic. Aliquots from this overnight grown culture was then used to prepare DMSO stocks, extract DNA for copy number estimation and WGS, and MIC determination as described above.

### MIC measurements
To determine the MIC of the isolates used in this study, Etest strips (Biomerieux) for tobramycin, gentamicin, and tetracycline were used. Briefly, the overnight cultures for each isolate were diluted 1:20 in phosphate-buffered saline (PBS, 8 g/L NaCl, 0.2 g/L KCl. 1.44 g/L NA2HPO4, and 0.24 g/L KH2PO4), spread on MH agar plates using sterile cotton swabs, and the Etest strips for the appropriate antibiotic were applied to the middle of the plates. After incubating the plates at 37 °C for 18-24, the MIC was determined by visual inspection.

### Bacterial relative growth rate measurements
The growth rate for all the individual isolates used in this study was analyzed using Bioscreen C (Oy Growth Curves AB, Ltd). For all isolates used in this study, ~3-4 µl cells were scraped from the −80 °C bacterial stocks. They were then diluted 1:1000 in MH broth and 300 µl aliquots were used to inoculate Honeycomb plates in six biological replicates (n = 6). The cultures were allowed to grow for 24 hours at 37 °C. The optical densities at 600 nm (OD600) were measured and recorded every 4 min and the cultures were shaken between measurements. Relative growth rates of the mutants as compared to the parental strains were determined considering the OD600 values between 0.2 and 0.9 using the Bioscreen Analysis Tool (BAT) 2.0 software.

### Determination of gene copy number
DNA was extracted from the 340 µl aliquot cell pellets stored at −20 °C using bead beating. The cell pellets were first resuspended in 250 µl of Fast Lysis Buffer for Bacterial Lysis Solution (QIAGEN) and nuclease free water (Sigma-Aldrich) mixed in 1:1 ratio. The suspension was then transferred to 2 mL freezing tubes containing 0.75 g of acid-washed glass beads with a diameter of 212–300 µm (Sigma-Aldrich). This was followed by adding 250 µl of Phenol:Chloroform:Isoamyl Alcohol (25:24:1) (Sigma-Aldrich) and incubating the tubes on ice for 2–5 min. Bead beating was performed for two rounds using a FastPrep-24TM Classic Instrument (MP Biomedicals) for 20 s at 6.5 m/s. The tubes were then centrifuged and after removing the aqueous phase, 200 µl of Phenol:Chloroform:Isoamyl Alcohol was added. The tubes were centrifuged again and the aqueous phase containing the DNA was removed and stored in a new tube at −20 °C until used for ddPCR. The gene copy number was determined using QX200TM ddPCR system (Bio-Rad) according to the manufacturer's recommendations. The ddPCR reactions were prepared with 2X EvaGreen Supermix (Bio-Rad). Briefly, for each reaction, 12.5 µl of 2X EvaGreen Supermix, 1µl of HindIII-HF (NEB) (final concentration: 2 U/reaction), 0.25 µl of each forward and reverse primers (100 nm each) and DNA template was mixed together and the final volume was adjusted to 25 µl with nuclease-free water (primer sequences in Supplementary Table 7). The ddPCR reaction was performed in a 96-well qPCR plate (Bio-Rad) by first generating droplets in an Automated Droplet Generator (Bio-Rad). The droplets were amplified in a PCR machine as follows: hot-start/enzyme activation at 95 °C for 5 min, 39 cycles of denaturation at 94 °C for 30 s and elongation at 56 °C for 1 min, followed by signal

stabilization at 4 °C for 10 min and 90 °C for 5 min and then cool down at 12 °C. For all steps, a ramp rate of 2 °C/s was used. Finally, the amplified droplets were read by a QX200 Droplet Reader and the data was analyzed using QuantaSoftTM software (Bio-Rad). As the changes in copy numbers were considerable and variability between repeat ddPCR runs is generally very small (as shown by control experiments with a few strains), we used only 1 technical replicate per strain and condition.

### Whole genome sequencing and analysis

The bacterial isolates were stored as cell pellets at −20 °C for DNA extraction and subsequent WGS. Genomic DNA extraction was performed using the Lucigen MasterPure Complete DNA and RNA Purifications Kit (Epicenter) according to the manufacturer's protocol. The extracted DNA was quantified using a Qubit 2.0 fluorometer (Invitrogen) and its purity was assessed using the Nanodrop 1000 (Thermo Fisher Scientific). Genomic DNA library preparation and sequencing were performed by BGI Genomics (Poland) using a DNBSeqTM platform. The sequencing files were then analyzed using CLC Genomics Workbench (version 11.0) to identify the point mutations and indels. Large deletions were identified by analyzing the sequencing files using the *breseq* pipeline[46].

### Strains construction

Select SNPs identified in known resistance genes and in genes with no known role in resistance were reconstructed in *E. coli* and *S.* Typhimurium lab strains using λ-red recombination[47]. Briefly, the lab strains carrying the pSIM5-tet plasmid were grown overnight in 2 mL LB (10 g/L NaCl, 10 g/L tryptone, 5 g/L and yeast extract) supplemented with 10 mg/L tetracycline at 200 rpm and 30 °C. 500 μl of these overnight grown strains were used to inoculate 50 mL of fresh LB supplemented with 10 mg/L tetracycline. The cultures were allowed to grow at 200 rpm and 30 °C until the OD600 reached 0.2–0.3. Expression of the λ-red genes on the plasmid were induced by incubating the cell cultures in a 42 °C water bath. The cells were shaken at 150 rpm for 20 min before cooling them on ice for 15 min. The cells were then washed three times by suspending them in ice-cold glycerol (10%) and centrifuging at 2564 x g and then suspended in 200 μl ice-cold glycerol (10%). A *cat-sacB*-YFP cassette[48] was previously PCR amplified with overhangs homologous to the chromosomal gene/region of interest (primer sequences in Supplementary Table 7). The selection cassette DNA was mixed with 40 μl electrocompetent cells and electroporation was performed at 1.8 kV, 200 Ω, and 25 μF. Immediately after electroporation, 0.5 mL of prewarmed LB (37 °C) was added to the cells followed by 3-4 min incubation at 42 °C. The cells were then mixed with 5 mL fresh LB and allowed to recover overnight at 30 °C and 190 rpm. 100 μl cells were plated out on LB agar plates containing 12.5 mg/L chloramphenicol. After confirming the insertion of the selection marker, the cells were transformed again with single stranded oligonucleotide containing the desired mutation (oligonucleotide sequences in Supplementary Table 7) following the same steps mentioned above. The transformants were selected on sucrose plates and insertion of the presence of the desired mutation were confirmed by Sanger sequencing.

### Stability test

The stability of the gene amplifications in the compensated mutants was determined and compared to the mutants isolated at 24X MIC of the susceptible population. These mutants were stored at −80 °C and serially passaged for 150 generations in the absence of antibiotics. Passages were done once every day by transferring 2 μl culture into 2 mL of fresh MH media (1000-fold dilution). At generation 20, 50, 100, and 150, an aliquot of the cell culture was stored at −80 °C in 10% dimethylsulfoxide (DMSO). In parallel, aliquots were also stored at −20 °C to extract DNA and determine the changes in gene copy

number. Additionally, the same culture was also used to determine the strain MIC using Etests for the respective antibiotic.

### Statistics and reproducibility

The statistical tests were performed in R (R Core Team 2022). The significance testing of the changes in relative growth rate (arbitrary cutoff: 5%) was done using unpaired two-tailed Student's *t* test. The goodness-of-fit was assessed using R-squared in regression analysis between the experimental data and the corresponding values obtained from the recombination model.

### Gene amplification model to theoretically determine recombination rate and fitness cost

A homologous recombination model[31] was used to theoretically determine the dynamics of gene loss. The model was used to determine the recombination rate ($k_{rec}$) and the cost of carrying each extra gene copies ($s$)[13,31,32]. This model assumes that recombination rate is higher for population containing higher gene copies and the presence of extra gene copies incurs a physiological cost reflected as reduced growth rate. The loss rate of gene copies in the absence of selection pressure in the compensated mutants as well as in their high copy number parental isolates were experimentally measured. The model was then simulated by varying the parameter values of $k_{rec}$ and $s$ to best mimic the observed loss rate dynamics from our experimental data. It should be noted that the behavior of the model is not dependent on the exact values of the parameters. As long as the assumptions of the model hold true, the model is robust to changes in the parameter values.

The main parameter affecting the frequency of gene amplifications in a population is the rate of homologous recombination, $k_{rec}$, that can lead to either an increase or decrease in copy number[13,31,32]. Assuming that all the gene copies are identical for recombination and $k_{rec}$ is constant for each gene copy, the overall probability of recombination per replication ($\rho^{rec}$) will increase with the number of copies present ($m$) as:

$$\rho_m^{rec} = \frac{k_{rec}(m-1)}{1+k_{rec}(m-1)} \tag{1}$$

This assumption is based on the consideration that cells with higher initial copy number have a higher chance to undergo recombination. After one recombination event, the probability of going from initial copy number $m$ to new copy number $n$ is given by:

$$\rho(n|m) = \frac{2m-n}{m^2} \text{ if } m<n \tag{2}$$

$$\rho(n|m) = \frac{n}{m^2} \text{ if } n < m \tag{3}$$

The probability goes to zero if $2m < n$ since it is not possible to get amplifications higher than that present during cell division. Additionally, it is assumed that higher number of gene amplifications will incur a higher fitness cost. Thus, for each extra copy of the gene, the cell's growth rate is reduced by the factor $(1+s)$ given that $s < 0$. Therefore, the fitness of the population with $m$ copies compared to the wild-type population carrying only one gene copy is given by:

$$w_m = (1+s)^{m-1} \tag{4}$$

In a cell population, several subpopulations exist wherein they carry varying number of gene amplifications. Each subpopulation competes based on their growth rates and the fitness of a given population is derived from the fitness contributions of each subpopulation carrying different number of gene copies. Assuming that a subpopulation with varying number of amplification ($n$) is present in fraction $f_n$ of

the population, the average fitness of the population at that time is given by:

$$W = \sum_n w_n f_n \qquad (5)$$

In each generation of the population, each of the fraction will change by:

$$\Delta f_n = \left(\frac{w_n}{W} - 1\right) f_n + \sum_{m=2}^{\infty} \rho_m^{rec} \rho(n|m) \frac{w_m}{W} f_m - \rho_n^{rec} \frac{w_n}{W} f_n \qquad (6)$$

Thus, the average number of gene copies after each generation will be given by:

$$<n> = \sum_n n f_n \qquad (7)$$

This model is fully deterministic and is based on the following assumptions:

1. The population is very large and the changes in gene copies are fast.
2. Recombination events can either lead to an increase in gene copy number or a decrease with equal probability. The loss of gene amplifications is therefore driven by the fitness cost which in turn limits the number of copies in a population and the summation to infinity in Eq. (6) can be restricted to a suitable upper value.

**Reporting summary**
Further information on research design is available in the Nature Portfolio Reporting Summary linked to this article.

## Data availability
The authors declare that all data supporting the findings of this study are available within the paper and its supplementary information files. Source data are provided with this paper. The WGS data generated in this study have been deposited in NCBI BioProject under accession number PRJNA1077788. Source data are provided with this paper.

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

## Acknowledgements
This work was supported by grants to DIA from NIH (grant 1U19AI158080-01), Swedish research Council (grant 2021-02091), and Wallenberg Foundation (grant 2018.0168). We thank Arianne Babina for her critical reading and comments.

## Author contributions
Conceptualization: Ankita Pal, Dan I. Andersson. Formal analysis: Ankita Pal, Dan I. Andersson. Investigation: Ankita Pal. Methodology: Ankita Pal. Resources: Dan I. Andersson. Supervision: Dan I. Andersson. Writing – original draft: Ankita Pal. Writing – review and editing: Ankita Pal, Dan I. Andersson.

## Funding

## Competing interests
The authors declare no competing interests.
