## [Peer Review File · Nature Communications]

Bacteria can compensate the fitness costs of amplified resistance genes via a bypass mechanismReviewer #1 (Remarks to the Author):

In this manuscript by Pal and Andersson, the authors investigate how fitness costs associated with resistance through tandem amplification of a resistance gene are ameliorated. To do this, the authors first identified resistant mutants by serially streaking single colonies on agar plates with increasing concentration of antibiotics. Next, they quantified the amplification of the resistance gene copy number using ddPCR, revealing a rapid increase seemingly correlated with the antibiotic concentration. This increase resulted concomitantly with a decrease in fitness (determined by the decrease in growth rate in the absence of antibiotic). Next, the authors evolved the isolated mutants at 24x MIC for 100 generations to investigate to what extent the fitness cost can be reduced. This revealed that most lines significantly reduced their resistance gene copy number and thus the associated fitness costs. WGS revealed that these mutants carried mostly known resistance-conferring mutations that allowed growth at high antibiotic concentrations while permitting the reduction of resistance gene amplification.

I am somewhat confused by the title of this paper, which focusses on the reduction of fitness costs of antibiotic heteroresistance. Heteroresistance typically refers to the presence of a subpopulation of resistant mutants. While this subpopulation can incur a cost if it's sufficiently large and exhibits a decreased growth rate — thus impacting the growth rate of the entire heteroresistant population — this specific scenario isn't the focus of the investigation here. Instead, the authors explore how mutants, resistant through the amplification of a specific resistance gene, compensate for their fitness costs. In this context, invoking the term "heteroresistance" might not be crucial to the core findings, and the title might benefit from a more precise reflection of the study's focus on the compensation of fitness costs by mutants with amplified resistance genes.

Were the authors able to pick up the resistance gene amplifications from the whole genome sequencing? I would expect that 20x-80x copy number increases would be easy to detect by analyzing the mapped sequencing reads, however, these are not indicated in Supplementary Table 3. Is this because all sequenced mutants completely reverted back to the WT copy number? At least the sequenced high copy number parental isolates should show this amplification clearly. Did the later show any other mutations besides gene amplification when compared to the original WT lines. Their genotype compared to the WT is not available.

Regarding Fig. 2C, a scatter plot might be more suitable than a line plot. Additionally, to establish a conclusive relationship between growth rate and copy number, it is advisable to conduct statistical testing.

One notable aspect is the absence of statistical testing throughout the manuscript. No statistical tests were performed to assess the significance of observed patterns.

Fig. 3 proved challenging to interpret. In the legend, the authors indicate that the lighter shade represents high copy number mutants, while the darker shade represents its three independent lineages. However, it seems that the darker shade might actually represent the ancestral line. I would like this to be clearer.

Why was there no replication for the copy number experiments? Are there specific experimental or biological concerns preventing replication?

In their discussion, the authors suggest that the absence of fitness reduction up to a 40-fold increase in copy number might imply a threshold effect. However, based on their own experiment determining the loss rate in the absence of selection, it seems more plausible to argue against this. A more compelling explanation is that using growth rate as a proxy for fitness may not be sensitive enough to detect the fitness cost of copy number increases below 40. Additionally, the loss rate of copy number appears to be non-linear, with increases at higher copy numbers being relatively more costly than similar increases at low copy numbers (as observed in PMID 16049495).

Reviewer #2 (Remarks to the Author):

In the manuscript 'Reducing the fitness costs of antibiotic heteroresistance via a bypass mechanism,' the authors combine experimental evolution and mathematical modeling to address a significant problem in antibiotic resistance. Heteroresistance represents a growing concern in public health, being a critical aspect of bacterial survival and adaptation, particularly in the context

of antibiotic resistance, and it occurs when a genetically homogeneous population exhibits a range of susceptibilities to an antibiotic, often produced by gene amplifications. The authors use experimental evolution to study the mechanisms by which bacteria can reduce the fitness costs associated with these amplifications. Their research reveals that bypass mutations can play a significant role in compensating for these costs, thereby facilitating the persistence and evolution of resistant strains. This finding is particularly important as it highlights a potential pathway through which bacteria can sustain high levels of resistance without imposing a high fitness cost, thus improving the stability of resistance mechanisms in the absence of selection.

While the study provides valuable insights into the evolutionary dynamics of heteroresistance, the mathematical model it employs requires further elaboration and detail. The model is based on a previously published model and aims to quantify the dynamics of gene amplification loss and the balancing of fitness costs. However, the presentation lacks specificity in how the parameters were determined and what do they mean in the context of the experimental data. To clarify these aspects and strengthen the manuscript, the authors should provide a more thorough explanation of the methods used to derive these parameters, including a measure of the model's goodness-of-fit with the experimental data. Additionally, a clearer and more informative presentation of the model's results would be beneficial. The caption of Supplementary Figure 4 indicates that various shades of darkness represent separate lineages from the compensatory evolution experiment, but these different shades are not visible in the figure.

I also believe the manuscript's title may lead to some confusion. Initially, it suggested to me that the study might propose a method for humans to actively reduce the fitness costs of antibiotic heteroresistance through a bypass mechanism. In reality, the manuscript focuses on how bacteria naturally evolve to offset these costs. This distinction is crucial and should be made clear to avoid any misunderstanding about the study's content and objectives and therefore I recommend the authors to revise the title.

In summary, this is a scientifically sound and thought-provoking study that has the potential to significantly advance our understanding of drug resistance evolution. The results presented provide key insights into the mechanisms by which bacteria mitigate the fitness costs associated with gene amplifications, a fundamental aspect of heteroresistance. However, a more detailed exposition of the model, along with a clearer articulation of its objectives and implications, would greatly enhance the understanding of its role in the broader context of the study. By addressing these aspects, the model could provide a more robust framework for interpreting the experimental data and predicting the evolutionary outcomes of antibiotic resistance, thereby enhancing the overall quality and impact of the manuscript.

Response to reviewer comments

Reviewer #1 (Remarks to the Author):

In this manuscript by Pal and Andersson, the authors investigate how fitness costs associated with resistance through tandem amplification of a resistance gene are ameliorated. To do this, the authors first identified resistant mutants by serially streaking single colonies on agar plates with increasing concentration of antibiotics. Next, they quantified the amplification of the resistance gene copy number using ddPCR, revealing a rapid increase seemingly correlated with the antibiotic concentration. This increase resulted concomitantly with a decrease in fitness (determined by the decrease in growth rate in the absence of antibiotic). Next, the authors evolved the isolated mutants at 24x MIC for 100 generations to investigate to what extent the fitness cost can be reduced. This revealed that most lines significantly reduced their resistance gene copy number and thus the associated fitness costs. WGS revealed that these mutants carried mostly known resistance-conferring mutations that allowed growth at high antibiotic concentrations while permitting the reduction of resistance gene amplification.

I am somewhat confused by the title of this paper, which focusses on the reduction of fitness costs of antibiotic heteroresistance. Heteroresistance typically refers to the presence of a subpopulation of resistant mutants. While this subpopulation can incur a cost if it's sufficiently large and exhibits a decreased growth rate — thus impacting the growth rate of the entire heteroresistant population — this specific scenario isn't the focus of the investigation here. Instead, the authors explore how mutants, resistant through the amplification of a specific resistance gene, compensate for their fitness costs. In this context, invoking the term "heteroresistance" might not be crucial to the core findings, and the title might benefit from a more precise reflection of the study's focus on the compensation of fitness costs by mutants with amplified resistance genes.

We agree with the reviewer's comment. We have therefore changed the title of the manuscript to "Bacteria can compensate the fitness costs of amplified genes via a bypass mechanism" which we think better reflects the findings of this study (Lines 1-2).

Were the authors able to pick up the resistance gene amplifications from the whole genome sequencing? I would expect that 20x-80x copy number increases would be easy to detect by analyzing the mapped sequencing reads, however, these are not indicated in Supplementary Table 3. Is this because all sequences compensated mutants completely reverted back to the WT copy number? At least the sequenced high copy number parental isolates should show this amplification clearly. Did the later show any other mutations besides gene amplification when compared to the original WT lines. Their genotype compared to the WT is not available.

Yes, we could indeed see the amplification of the antibiotic resistance gene from our whole genome sequencing data for the high copy number parental isolates as well as the compensated mutants by comparing the coverage of the mapped sequencing reads. Since WGS could give only a relative estimate of the changes in amplification levels, we chose to mention the amplification levels from our ddPCR data wherein we could measure the absolute copy numbers of the resistance genes with very high precision.

Also, the high copy number parental isolates did not have any other mutations apart from the gene amplifications and therefore they were not mentioned in Supplementary Table 3. We have now mentioned this explicitly in the manuscript (Lines 201-205).

Regarding Fig. 2C, a scatter plot might be more suitable than a line plot. Additionally, to establish a conclusive relationship between growth rate and copy number, it is advisable to conduct statistical testing.

We think the line plot better reflects our data as the lines show the 4 different lineages from our experiment.

Our data shown in Fig. 2C shows that there is no linear correlation between growth rate and copy number. We did not observe any drastic changes in relative growth rate for mutants carrying up to 40 copies. For mutants carrying substantially higher copies of gene

amplification (60-70 copies), we could observe a significant fitness defect (Line 128). This indicates a threshold effect of copy number for reasons that are not clear yet.

One notable aspect is the absence of statistical testing throughout the manuscript. No statistical tests were performed to assess the significance of observed patterns.

This is an oversight on our part and we thank the reviewer for pointing this out. We have performed statistical testing wherever necessary and have included that in the manuscript (Lines 128, 153, 217), figure 3 and supplementary figures 1, 2, and 4. We have also added a section about statistical tests in Mat-Met (lines 535-541).

Fig. 3 proved challenging to interpret. In the legend, the authors indicate that the lighter shade represents high copy number mutants, while the darker shade represents its three independent lineages. However, it seems that the darker shade might actually represent the ancestral line. I would like this to be clearer.

We apologize for the mistake. We have rectified the legend.

Why was there no replication for the copy number experiments? Are there specific experimental or biological concerns preventing replication?

We apologize but we misstated this. There are 3 biological replicates (each lineage in fig. 2 and 3) and one technical replicate. This has been corrected in the legends for figs. 2 and 3. We usually do only one technical replicate when using ddPCR since the technical variability is very low.

In their discussion, the authors suggest that the absence of fitness reduction up to a 40-fold increase in copy number might imply a threshold effect. However, based on their own experiment determining the loss rate in the absence of selection, it seems more plausible to argue against this. A more compelling explanation is that using growth rate as a proxy for fitness may not be sensitive enough to detect the fitness cost of copy number increases below 40.

Additionally, the loss rate of copy number appears to be non-linear, with increases at higher copy numbers being relatively more costly than similar increases at low copy numbers (as observed in PMID 16049495).

We would like to argue that even though our method to measure relative growth rates might not be highly sensitive, we are able to measure relative growth rates with no more than 5% error. With the number of replicates used in the growth rate experiments, we observed very low standard deviations in our data, and we believe that this indicates that there is indeed a threshold effect of copy number increases on the fitness of the organism.

Reviewer #2 (Remarks to the Author):

In the manuscript 'Reducing the fitness costs of antibiotic heteroresistance via a bypass mechanism,' the authors combine experimental evolution and mathematical modeling to address a significant problem in antibiotic resistance. Heteroresistance represents a growing concern in public health, being a critical aspect of bacterial survival and adaptation, particularly in the context of antibiotic resistance, and it occurs when a genetically homogeneous population exhibits a range of susceptibilities to an antibiotic, often produced by gene amplifications. The authors use experimental evolution to study the mechanisms by which bacteria can reduce the fitness costs associated with these amplifications. Their research reveals that bypass mutations can play a significant role in compensating for these costs, thereby facilitating the persistence and evolution of resistant strains. This finding is particularly important as it highlights a potential pathway through which bacteria can sustain high levels of resistance without imposing a high fitness cost, thus improving the stability of resistance mechanisms in the absence of selection.

We thank the reviewer for the positive comments.

While the study provides valuable insights into the evolutionary dynamics of heteroresistance, the mathematical model it employs requires further elaboration and detail. The model is based on a previously published model and aims to quantify the dynamics of gene amplification loss and the balancing of fitness costs. However, the presentation lacks specificity in how the parameters

were determined and what do they mean in the context of the experimental data. To clarify these aspects and strengthen the manuscript, the authors should provide a more thorough explanation of the methods used to derive these parameters, including a measure of the model's goodness-of-fit with the experimental data. Additionally, a clearer and more informative presentation of the model's results would be beneficial.

We agree with the reviewer's comments. Thus, we have included more detail regarding estimation of the parameter values in the Materials and Methods section (Lines 485-495). We have also assessed the goodness-of-fit for the model with the corresponding experimental data and have included that in the Supplementary Fig 4. We have also explained the results of the model in more detail in the Results section (Lines 264-268 and 271-275).

The caption of Supplementary Figure 4 indicates that various shades of darkness represent separate lineages from the compensatory evolution experiment, but these different shades are not visible in the figure.

We have modified the colors in the figure to make the distinction between the parental high copy number isolates and the compensated mutants clearer.

I also believe the manuscript's title may lead to some confusion. Initially, it suggested to me that the study might propose a method for humans to actively reduce the fitness costs of antibiotic heteroresistance through a bypass mechanism. In reality, the manuscript focuses on how bacteria naturally evolve to offset these costs. This distinction is crucial and should be made clear to avoid any misunderstanding about the study's content and objectives and therefore I recommend the authors to revise the title.

We have revised the title of the manuscript to "Bacteria can compensate the fitness costs of amplified genes via a bypass mechanism" to avoid confusion (Lines 1-2).

In summary, this is a scientifically sound and thought-provoking study that has the potential to

significantly advance our understanding of drug resistance evolution. The results presented provide key insights into the mechanisms by which bacteria mitigate the fitness costs associated with gene amplifications, a fundamental aspect of heteroresistance.

We thank the reviewer for the positive comments.

However, a more detailed exposition of the model, along with a clearer articulation of its objectives and implications, would greatly enhance the understanding of its role in the broader context of the study. By addressing these aspects, the model could provide a more robust framework for interpreting the experimental data and predicting the evolutionary outcomes of antibiotic resistance, thereby enhancing the overall quality and impact of the manuscript.

We agree with this and we have tried to clarify and develop the implications of our findings. We have outlined the objectives (Lines 245-251) and the implications (Lines 337-339) in the manuscript. In addition, we have added a more general conclusion from our findings to indicate that this type of compensatory process to reduce costs of gene amplifications is likely to occur in other cases as well (Lines 348-353).

Reviewer #1 (Remarks to the Author):

I have no additional comments. The authors have satisfactorily addressed my feedback, and I recommend accepting their article. Congratulations to them on their excellent work!

Reviewer #2 (Remarks to the Author):

After reviewing the revised manuscript, I am pleased with the authors' revisions. They have addressed my concerns effectively, notably enhancing the manuscript's clarity. The modifications to the mathematical model, the improved figures, and the more appropriate title collectively make the paper ready for publication.